# Epidemiological Trends of Trans-Boundary Tick-Borne Encephalitis in Europe, 2000–2019

**DOI:** 10.3390/pathogens11060704

**Published:** 2022-06-18

**Authors:** Mulugeta A. Wondim, Piotr Czupryna, Sławomir Pancewicz, Ewelina Kruszewska, Monika Groth, Anna Moniuszko-Malinowska

**Affiliations:** Department of Infectious Diseases and Neuroinfections, Medical University of Bialystok, Żurawia 14 Block E, 15-540 Białystok, Poland; avalon-5@wp.pl (P.C.); spancewicz@interia.pl (S.P.); kruszewska.ewelina@gmail.com (E.K.); mkrol94@gmail.com (M.G.); annamoniuszko@op.pl (A.M.-M.)

**Keywords:** tick-borne encephalitis, tick-borne encephalitis virus, trend, climate change, ticks

## Abstract

Tick-borne encephalitis is a neuroinfection widely distributed in the Euro–Asia region. Primarily, the virus is transmitted by the bite of infected ticks. From 2000–2019, the total number of confirmed cases in Europe reported to the European Centre for Disease Prevention and Control was 51,519. The number of cases decreased in 2014 and 2015; however, since 2015, a growing number of cases have been observed, with the involvement of countries in which TBE has not been previously reported. The determinant factors for the spread of TBE are host population size, weather conditions, movement of hosts, and local regulations on the socioeconomic dynamics of the local and travelling people around the foci areas. The mean incidence rate of tick-borne encephalitis from 2000–2019 in Europe was 3.27, while the age-adjusted mean incidence rate was 2.19 per 100,000 population size. This review used several articles and data sources from the European Centre for Diseases Prevention and Control.

## 1. Introduction 

Tick-borne encephalitis (TBE) is a viral infectious disease caused by tick-borne encephalitis virus (TBEV) that affects the central nervous system. It appears in mild, moderate, or severe forms, with the possibility of long-lasting neurologic sequelae development. It is mainly transmitted by an infected tick bite and rarely through the ingestion of raw milk [1]. A report has also revealed that three solid organ transplant recipients were infected with TBEV during transplantation [2]. 

The transmission of the virus occurs between ticks, animals, and humans. Humans are the dead-end host of TBEV [3]. In Europe, the *Ixodes ricinus (IR)* tick is the main vector for the transmission of TBEV [4]. *IR* ticks are prevalent throughout Europe, from Ireland in the west to the Urals in the east, and from northern Sweden to north Africa [5]. Birds, sheep, goats, horses, rodents, dogs, and other animals can also host the virus, as well as humans [6]. Animals considered to be suitable hosts remain infected with sufficient amounts of the virus in their blood stream to infect naïve ticks while they are feeding. Co-feeding is the other way that ticks get infected. Simultaneous feeding by both infected and non-infected ticks in proximity leads the uninfected ticks to acquire the virus [7,8]. Besides transmission from host to vector or vice versa, the transovarian transmission of TBEV from parent to offspring and transstadial transmission helps the virus to pass through all developmental stages [7,9]. Even though ticks have their own preference for feeding on animals during a particular developmental stage, small mammals such as rodents and insectivores are believed to be their main hosts. Small- and medium-sized mammals and birds are the targets of larvae and nymphs, while adults stick to infesting larger animals [7,9,10]. 

There are three common subtypes of TBEV, known as European (TBEV-Eu), Siberian (TBEV-Sib), and Far-Eastern (TBEV-FE). *IR* (TBEV-Eu) and *Ixodes persulcatus* ticks carry these three antigenically closely related subtypes [3]. Recently, two more subtypes, Baikalian (TBEV-Blk) and Himalayan (TBEV-Him), have been discovered [11,12]. TBEV-Eu causes neurological sequelae in up to 10% of cases, with a mortality rate of < 2% [13,14,15]. TBEV-Sib causes prolonged infection compared to other sub-types, while TBEV-FE patients show high rates of neurological sequelae [13]. 

TBE was discovered in 1930 by scientists from the former Union of Soviet Socialist Republics (USSR). The first subtype discovered was the Far-Eastern subtype. There was confusion between TBE and Japanese encephalitis until scientists confirmed that they were different and that the route of the transmission of TBE was not airborne [16]. The Western subtype was reported as seasonal meningitis from Austria [17]. 

Ticks have a long life cycle, and TBEV can survive throughout their developmental stages. *Ixodid* ticks ingest blood once in a developmental stage, and they have no chance of transmitting TBEV until they feed again [18,19]. The life cycle of ticks is affected by the microclimate, host factors, and seasonal variation. During colder seasons, the activity and development of the two common species of ticks are limited [4]. Ticks become active during vegetation seasons, with adequate levels of humidity and elevated temperatures. Access to increased moisture help ticks ascend to the upper part of grasses, thereby increasing their opportunities to attach to hosts. During molting time, the size of ticks shrinks with the release of water and the hardening of the skin. At this time, ticks prepare for the winter season until the following spring [20]. 

Globally, TBE covers the geographical areas of Japan, China, Russia, Southern Europe, Central Europe, and Northern Europe. In Europe, the most strongly affected countries are southern Germany, Switzerland, the Czech Republic, Austria, Slovakia, Hungary, the Baltic countries, Slovenia, Poland, parts of Scandinavia, and European Russia [3]. The recent presence of TBEV in Northern Europe is evidence for the emergence of new TBE foci [21].

## 2. Factors Affecting TBEV Transmission 

The risk of TBE in humans is influenced by, e.g., socioeconomic changes, climate change, seasonal variations, and individual predispositions. An extended and warmer summer and the increased population size of hosts favor the survival of ticks and microbes outside of their known habitats [22,23,24]. After an extended and warmer summer, the dynamics of the other seasons change in a way that is favorable for ticks’ survival [24]. Mild and shorter winters are followed by rainy summers, which help grass grow effectively so that ticks can climb and infest humans and other hosts. It has been observed that ticks remain active between March and December because of climate change, which causes shorter winters. The duration of the convenient season for tick activity extended after the year 2000 [25]. Climate change affects the survival, reproduction, interaction with hosts and the environment, and movement of ticks. Cases have been reported from new areas because of activities related to increased human need and the possibility of accessing places that were hard to reach in the past [26]. 

During winter, ticks hide in the leaf litter or soil and digest the blood ingested during their questing time. Ticks require warm temperatures and humidity to become active (exiting from the soil and ascending into the vegetation) [26,27]. Ticks can be active in dry seasons, looking for hosts and food; however, such conditions limit their longevity [28]. Weather conditions with temperatures of around 8°C and humidity levels of 70–80% are ideal for tick movement and feeding. The European subtype (adult *IR* tick) is most active in the periods of May–June and September–October [21]. 

Although climate change is a major factor in the increased spread of several infectious diseases, including TBE, in humans, socioeconomic changes play a comparable role. The end of the Soviet era caused a surge in the number of TBE cases in humans within countries that had been influenced by the planned economy. This was related to the decreased pesticide use, the expansion of subsistence farming and land use, and increased leisure time [29]. Reduced pesticide use favors the survival of ticks and rodents, seemingly promoting TBEV transmission. Small-farm owners tend to fail at following safety protocols, and so people consume dairy products which could be infected with TBEV. In Lithuania, the incidence rate in a joinpoint model rose by 195.2% after 1991 [30]. Estonia also saw a sharp rise in TBE cases in the 1990s. From 1976 to 1992, the yearly incidence rate per 100,000 was 3, while 12, 28, and 27 cases were reported in 1996, 1997, and 1998, respectively [31]. In the Czech Republic, an increase in the incidence rate of TBE was observed in the years 1990–2008 compared to 1970-1989 (*p* < 0.05) [32]. 

The number of human cases reported to the ECDC shows seasonal variation and a bimodal distribution, with the peak number of cases reported between June and August, followed by the second peak in October. The majority (95%) of cases are reported from May to November [33]. The recent increased incidence rate is also associated with the advancement of diagnosis tools and the establishment of mandatory reporting and surveillance systems; in the past, a considerable proportion of cases remained either undetected or unreported [34]. However, the irregular geographical distribution with patchy foci and the low prevalence of TBEV in both hosts and vectors make controlling TBE difficult [35,36]. 

The population of ticks depends on the density of hosts, though not all hosts are able to transmit the virus. The composition of hosts in a specific area determines the impact of hosts on tick survival. The higher proportion of non-competent hosts reduces the transmission of TBEV by minimizing the number of tick bites [37]. Changing the land use and cover also increases the accessibility of landscapes for humans. Access to tick-populated areas by humans eventually leads to an increased chance of acquiring tick bites [37]. Converting agricultural fields to forests creates a convenient environment for hosts’ and ticks’ survival [38]. 

The exposure to ticks is also increased with the movement of people [25]. Human mobility in tick-populated areas is a potential factor for the enhanced spread of both ticks and TBE. People travel to forest areas for activities such as hiking and mushroom or berry picking. In 2007 alone, 78 million people travelled to TBE endemic areas in Europe without knowledge of the disease [39]. 

## 3. Migration of TBE Subtypes

In recent decades, the European strain of TBEV has spread into non-endemic areas. A phylogenetic analysis conducted in Hungary (2011–2016) revealed that Hungarian strains formed a unique cluster that showed similarities to TBEV strains from Finland, Germany, and Russia. Trans-boundary migratory birds were supposedly responsible for the movement of other countries’ strains [40]. The Siberian subtype was found in the northernmost part of Europe, including Finland [41] and Estonia, and three of the subtypes were found in the Crimean Peninsula [42]. 

Evidence has emerged showing either the appearance of TBEV in new areas or the re-emergence of TBE. In the United Kingdom, the first occurrence of TBEV was reported in May 2019, when the virus was found in ticks collected from deer. The Netherlands saw its first human case in 2016 [12]. Denmark has reported cases in its Baltic regions [43]. In northern Germany, ticks tested positive for TBEV after 15 years of its absence from the area, either due to long-term persistence at low levels of activity or virus reappearance due to bird migrations [44]. The first autochthonous cases of TBE were reported in Moscow in 2016, along with the detection of TBEV in ticks and small mammals [45].

The altitude convenient for ticks in the past was limited to the lower part of Europe. However, other conditions impact tick population density regardless of the altitude level, such as proximity to the ocean, humidity level, and vegetation. In Europe, ticks are reported to survive at altitudes between 600 and 2000 m above sea level with other elements required for their survival (Table 1) [29,46]. Multiple combined factors contribute to an increase in the risk of TBE incidence. Recently, the popularity of mountain climbing as a recreational activity has increased, and ticks have adapted to new foci at higher altitudes because of climate-change-induced warmer weather [29,39,46]. 

Predictions show that the foci are spreading to the northern part of Europe, while the number of cases is decreasing in central Europe. The spread is expected to be limited to Poland and the Baltic area in 2050 and the southern part of Scandinavia by 2080. The predicted global land temperature and precipitation will not be favorable for the spread of TBEV in its human and enzootic hosts. The temperature change in the land will be higher than that in the sea, leading to less rainfall. Eventually, it is believed that the moisture level in central Europe will be insufficient for ticks [47,48]. 

**Table 1 pathogens-11-00704-t001:** Tick survival with altitude change over time, from 1950 to 2013.

Year	Country	Altitude (Meters above Sea Level)	Conditions
1950	Bosnia Herzegovina	<800	[49]
1957	Scotland	700	[49]
1960	Bosnia Herzegovina	900	[48,49]
1980	Czech Republic	700–750	Krkonose mountainous area [50]
1997	Italy	<1300	Presence of limestone and vegetation cover with thermophile deciduous forests and high densities of roe deer [51]
1990s	Scotland	≥700	[49,52]
2001	Scotland	1100	[49,52]
2002	Czech Republic	1180	Krkonose mountainous area [50]
2006	Czech Republic	1250	Krkonose mountainous area [50]
2008	Austria	>1500	Alpine pasture [53]
2008	Greece	>600	Arid parts of Europe typical of the Mediterranean
2010	Bosnia Herzegovina	1190	[49]
2013	Spain	2000	Atlantic influence (i.e., greater humidity) [49]
	Switzerland	1450	Mountainside [49]

Controlling TBEV is extremely difficult because of its unique life cycle, which spans arthropod vectors and reservoir hosts without involving humans [54]. TBE has no definitive treatment other than supportive treatment [55]. Therefore, vaccination plays an irreplaceable role [55,56]. In Austria, from 2000 to 2011, the overall field effectiveness in regularly vaccinated people was 96–99%. This reduced the number of TBE cases by over 4000 within the period 2000–2011 [54]. Two main TBE vaccines (Encepur and FSME-IMMUN) are available in Western Europe [57]. The first vaccine used in Europe was FSME-IMMUN^®^ (Pfizer, New York, NY, USA) in 1976, followed by Encepur^®^ (Bavarian Nordic) from Germany in 1991. Vaccines offer the best protection against TBEV. Vaccination is highly recommended for travellers who plan to stay in risky areas [56]. Post-exposure prophylaxis with specific anti-TBEV immunoglobulins is recommended in Russia and Kazakhstan; however, in Europe, its use has been discontinued because of concerns regarding antibody-mediated disease enhancement in naïve individuals [15].

## 4. Epidemiological Trends of TBE, 2000–2019

This review draws data from the European Centre for Disease Prevention and Control (ECDC) collected from European Union/European Economic Area member countries. The number of countries reporting TBE cases to the ECDC increased from 12 in 2000 to 25 in 2019. The data were gathered by ECDC surveillance (Table 2) and are used as per the data-use protocol of the ECDC. The database is classified into two parts: 2000-2010 and 2012–2019. The 2000–2010 section includes the total number of reported cases, total number of confirmed cases, and classification per sex per country. The 2012–2019 report includes the total number of cases, confirmed cases, confirmed cases per 100,000 population, and confirmed cases adjusted for age per 100,000 population. To show the historical background of TBE, several pieces of literature from diverse sources were included.

Over the past nineteen years among the European Union/European Economic Area member countries listed in Table 3, a total number of 51,519 confirmed TBE cases were reported to the ECDC. The mean incidence rate per year within the region was 3.26, and the mean age-adjusted incidence rate was 2.19 per 100,000 population per year. The number of countries included in the report has increased over the years. Lithuania, Latvia, and the Czech Republic had the highest incidence rate per year (13.66, 9.95, and 6.14 cases per 100,000 inhabitants per year, respectively). This, however, does not mean that there are no areas with a greater incidence rate (Figure 1). These countries are at the top since the report only considered their respective national figures.

However, in areas such as the eastern and northeastern regions of Poland, the incidence rate is closer to the national incidence rate of Lithuania. The number of cases between 2000 and 2013 was variable, increasing in some years and decreasing in others; however, the number of confirmed cases steadily increased from 2015 to 2019. Therefore, greater attention should be paid to foci areas, since the national reports of a particular country do not represent the uniform distribution of TBE cases across its regions.

The number of cases reported shows seasonal variation. The number rises in March, peaks in August, and declines in October (Figure 2).

TBE seems to be more prevalent among older adults in both sexes. Older people are more vulnerable to symptomatic infection and severe forms of the disease due to a decreased immune response. Vaccine failure is also more often observed among elderly people, as antibody response to TBEV vaccine declines with increased age (Figure 3) [58]. Furthermore, people aged >50 years are more likely than other age groups to travel across Europe without enough awareness about TBE and without protecting themselves against it. People in this age group are thus at a higher risk of TBE [39].

## 5. Conclusions

TBE is believed to be under-reported. The under-identification of cases could be related to a lack of clear-cut and specific diagnostic methods and the training of professionals. The dynamic identification of foci areas should be carried out to monitor the continuously changing expansion of TBE. An integrated public health intervention plan is also important to deter TBE expansion within Europe, including vaccination and disease detection, considering that TBE is an important disease affecting travellers and the general public.

## Figures and Tables

**Figure 1 pathogens-11-00704-f001:**
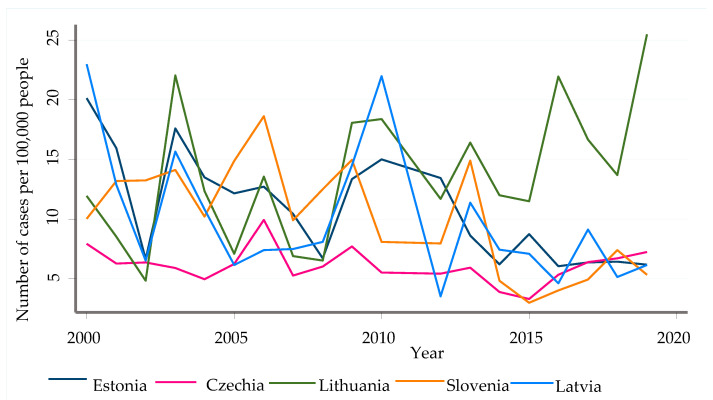
Change in the annual incidence rate of TBE in most affected countries (2000–2019).

**Figure 2 pathogens-11-00704-f002:**
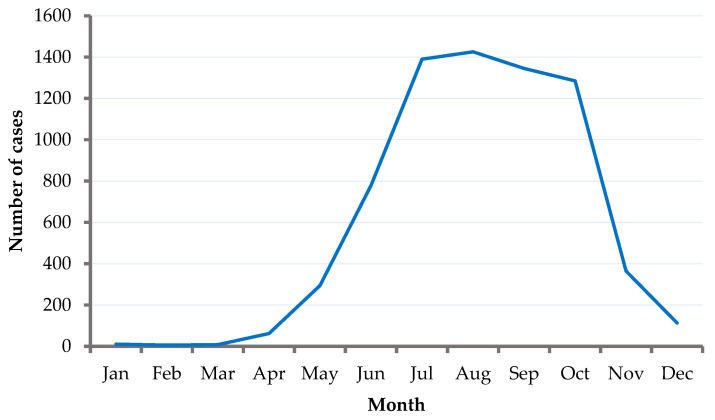
Number of TBE cases by month of onset reported in nine EU/EFTA countries (2000–2010; n = 7083). Source: data from the ECDC (https://register.ecdc.europa.eu/) (accessed on 19 January 2021).

**Figure 3 pathogens-11-00704-f003:**
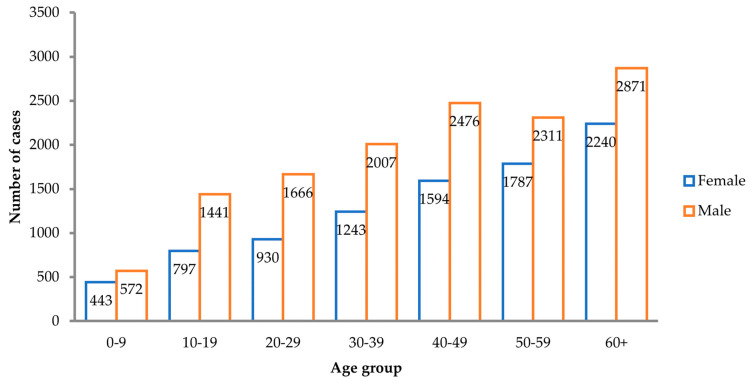
TBE Key risk groups: number of TBE cases by age group and gender reported in 16 EU/EFTA countries (2000–2010, n = 22,378). Source: data from the ECDC (https://register.ecdc.europa.eu/) (accessed on 19 January 2021).

**Table 2 pathogens-11-00704-t002:** The number of confirmed cases reported to the ECDC and number of countries involved each year (2000–2019).

Year	Number of Countries	Number of Confirmed Cases	Number of Confirmed Cases/100,000 Population
2000	12	2629	7.02
2001	14	2497	5.07
2002	14	1952	3.19
2003	14	3225	6.15
2004	14	2481	4.33
2005	15	2651	3.71
2006	15	3756	4.94
2007	15	2267	3.27
2008	16	2513	3.15
2009	16	3513	5.07
2010	16	3155	5.06
2012	21	2149	2.34
2013	22	2904	3.03
2014	24	1985	1.75
2015	24	1908	1.75
2016	25	2680	2.14
2017	25	2916	2.22
2018	26	3092	2.07
2019	25	3246	2.57
Total		51519	

Source: data from ECDC (https://register.ecdc.europa.eu/); (accessed on 16 January 2021).

**Table 3 pathogens-11-00704-t003:** Number of TBE cases, number of years the country has reported, and mean incidence rate (2000–2019).

Country	Number of Years	Confirmed Cases	IR/100,000	Age-Adjusted IR/100,000 (2012–2019)
Czechia	19	12055	6.14	5.50
Lithuania	19	8178	13.66	15.64
Germany	19	6089	0.41	0.43
Latvia	19	4184	9.95	6.60
Poland	19	3933	0.54	0.41
Slovenia	19	3877	10.12	6.34
Sweden	19	3345	2.48	2.85
Estonia	19	2761	10.86	7.73
Slovakia	19	1734	1.69	2.15
Austria	19	1562	0.97	1.10
Switzerland	11	1302	.	.
Hungary	19	878	0.46	0.24
Finland	19	787	0.77	1.07
Italy	18	312	0.03	0.03
Croatia	8	189	0.56	0.53
Norway	19	186	0.20	0.30
France	8	67	0.01	0.01
Romania	19	26	0.01	0.01
Netherlands	4	16	.	.
Belgium	8	14	0.02	0.02
United Kingdom	8	9	0.00	0.00
Denmark	1	4	0.07	0.07
Bulgaria	6	4	0.01	0.01
Greece	8	4	0.00	0.01
Luxembourg	6	1	0.03	0.03
Ireland	8	1	0.00	0.00
Spain	8	1	0.00	0.00
Total		51519	3.26 **	2.19 **

Source: data from the ECDC (https://register.ecdc.europa.eu/) (accessed on 16 January 2021) ** mean incidence rate. IR: incidence rate.

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
