# Peer review of "Epidemiological Trends of Trans-Boundary Tick-Borne Encephalitis in Europe, 2000–2019"

_pathogens, 2022, doi:10.3390/pathogens11060704_

Round 1

Reviewer 1 Report

The manuscript is nott easily readable. I suggest shortening and restructuring avoiding unnecessary repetitions, as row 94 and 170 "host the virus", 115 and 147 "ticks become active", r. 246-,373-,385- vaccination details. Just to name a few.

Addition of some explanatory details would be valuable. Difference in hosts - competent reservoir, support of tick reproduction or just getting infected. Role of co-feeding, predators? The vaccination rate in Poland and effect on the epidemiologic figures. Diagnostic principles in Poland and the relation to vaccination status.

The first sentence in conclusions is not clear.

Reviewer 2 Report

The review by Wondim et al received in the first round of the review process could have given a publishable article with a major (!) overhaul of the work. Unfortunately, the authors were either not willing or not capable of substantial improvements to their text. Despite having written “accepted” to some of the criticism on their paper in the first round no improvement in the overall quality of the manuscript can be seen in this modified version. The authors did not really address any of the major flaws in the first version and tried to respond to the criticism with the minimal amount of work possible. Unfortunately, this leads to a text that, while containing some sources with useful information, is in itself not able to concisely summarize the current state of TBEV epidemiological trends in Europe.

The text is full of…

  1. Poor presentation and interpretation of the information presented (only a few of several instances mentioned):
  • g. “non-infective” ticks (due to their biting only once at each stage) in Ref 14 transform to non-infective TBEV (inactive virus) in the text. Moreover, a paper concerned purely with the mathematical modeling of transmission of tick-borne diseases is the wrong kind of source for information on the molecular biology of said pathogen in its host.
  • Another example is the comment on the molecular time work cited in the text (“keep in mind that the method is erratic and has weak temporal signal”). Instead of taking the published data, discussing strength and weaknesses so that the reader might gains an estimation of the reliability of these estimates. Instead the text very much suggests that the authors themselves did not understand the cited paper.
  1. Very vague statements as for example
  • Line 153/154: “This relates to the decreased pesticide use, expansion of subsistence farming and land use and increased leisure time”. This compacts the information in the source in a way that the relation and interdependence between the factors can no longer be guessed at by the reader. Thus it becomes nearly meaningless.
  1. Shoddily written text (only a few of several instances mentioned):
  • g. comparing incidence rate (cases per 100 000) with cases (lines 159/160). I am guessing cases means cases per 100 000, as the sentence would be contradictory to the one before, but this is not what is written in the text.
  • Line 250/251 Meaningless sentence, because there is no context: This has reduced the number of TBE cases by over 4000 (cases? Percent? Yearly incidence? Total since start of the vaccination?)
  • Something is wrong with this sentence: “The determinant factors for the spread of TBE are host population size, weather conditions, movement of hosts, and local regulations on socio-economic dynamics of the local and travelling people around the foci areas.”
  1. Sloppy formatting and writing:
  • Author formatting not uniform
  • Change of font size in the text
  • ECDC abbreviation already used before full length name is introduced
  • Abstract contains deceased instead of decreased
  • Still poorly formatted bibliography in which several sources in the text only show error messages.

Reviewer 3 Report

Dear authors,

sincere compliments for your interesting work. 

Line 168 : Add a sentence to explain that TBEV control is difficult even because it showes a patchy distribution in small natural foci.

Line 173:  Check the font

Best regards

Reviewer 4 Report

Manuscript entitled “Epidemiological Trends of Trans-Boundary Tick-Borne Encephalitis in Europe, 2000-2019” by M.A. et al. describes epidemiology situation of TBE in Europe based on information from European Centre for Diseases Prevention and Control and Polish National Public Health Institute.

Main remarks:

  1. I miss clear opinion of authors on topic “Epidemiological Trends of TBE in Europe”. Does the number of cases increase or decrease in Europe in mentioned years? Authors describe in abstract only that the number of cases deceased in 2014-2015 and growing from 2015 and show total incidence rates.
  2. I miss any statistical analysis of data to make such statement about epidemiological trends. I can see in table 2, that 3756 and 3513 cases were reported in years 2006 and 2009 when 15 and 16 countries respectively were involved in analysis. On the contrary, 3246 cases were reported in year 2019 when 25 countries were included in the data acquisition. Therefore, in my opinion the reported cases per year should be standardised per population of countries where data were acquired. Also regression analysis should be done to show if the number of cases increase or decrease in Europe.
  3. The topic of article is “Epidemiological Trends of TBE in Europe” so there is no reason to show specific TBE Trend in Poland (chapter 5. starts at line 250). If the authors wants to publish these information it should be done in the form of independent article focused on situation in Poland. As it was done for example in Germany: Hellenbrand et al.: Epidemiology of Tick-Borne Encephalitis (TBE) in Germany, 2001–2018, Pathogens 2019. Moreover the statements about changes in TBE numbers in mentioned regions and spreading of TBE are not statistically analysed (lines 277-281).
  4. Figures should be presented in united format, now 4 graphs are presented in different format.

Minor remarks:

lines 4-18: The affiliation 1-6 differs only in e-mail address.

lines 57-62:The sentence should be reformulated and simplified.

line 64: I do not understand what you mean with .... of ticks by staying non-infective.

lines 174-175: bigger lettering.

figure 2: title of Y-axis is missing, source of data is missing. Authors start with figure 2, I recommend start with figure 1.

table 3: IR for Switzerland and Belgium is missing.

figure 4: No of cases is hard to read in age groups it overlap with columns.

figure 5: title of Y-axis is missing

Round 2

Reviewer 1 Report

The corrected manuscript can be published

Reviewer 4 Report

I am fine with  answers and corrections of authors.

The figures should be still presented in better format.

Figure 2 is presented twice in manuscript.

Figure 1: lines are very thin. Legend clould be presented just in one line without frame.

Figure 3: The columns should be presented as open bars just with coloured border - the number will be better visible.
